# Watermelon Nutritional Composition with a Focus on L-Citrulline and Its Cardioprotective Health Effects—A Narrative Review

**DOI:** 10.3390/nu17203221

**Published:** 2025-10-14

**Authors:** Diego dos Santos Baião, Davi V. T. da Silva, Vania M. F. Paschoalin

**Affiliations:** 1Postgraduate Program in Food Sciences, Department of Biochemistry, Chemistry Institute, Federal University of Rio de Janeiro (UFRJ), Rio de Janeiro 21941-909, RJ, Brazil; diegobaiao20@hotmail.com (D.d.S.B.); davivieiraufrj@gmail.com (D.V.T.d.S.); 2Postgraduate Program in Chemistry, Technology Center, Federal University of Rio de Janeiro (UFRJ), Rio de Janeiro 21941-909, RJ, Brazil

**Keywords:** *Citrullus lanatus*, L-arginine bioavailability, watermelon bioactive compounds, nitric oxide synthesis, cardiovascular risks management, cardiovascular health strategies

## Abstract

Nitric oxide is a gaseous molecule endogenously produced by endothelial cells, which stands out for its vascular tone regulation effects after crossing through the endothelium and diffusing to smooth blood vessel muscle cells. Reduced nitric oxide bioavailability contributes to the development of hypertension, atherosclerosis, worsening endothelial function, arterial stiffness, and ineffective stimulation of smooth muscle relaxation. L-citrulline, an amino acid found in high concentrations in watermelon, may serve as a recycling substrate, increasing L-arginine availability and, consequently, nitric oxide synthesis. By enhancing circulating L-arginine, L-citrulline indirectly improves the synthesis and bioavailability of nitric oxide, promoting smooth muscle vasodilation. Herein, this narrative review critically examines current evidence of the cardiovascular benefits of L-citrulline ingestion obtained exclusively through watermelon consumption, exploring the nutritional and bioactive composition of the edible parts of this fruit and the metabolism and effects of L-citrulline supplementation on vascular and metabolic physiology and proposing directions for future research, such as long-term studies and studies in specific populations. The beneficial effects of oral L-citrulline ingestion through watermelon require additional evidence, but it has already been demonstrated that it does not undergo hepatic metabolism, instead being transported to the kidneys to participate in de novo L-arginine synthesis. The generation of endogenous NO then causes positive biochemical, hemodynamic, and vascular effects, remodeling the physio-pathological conditions of those adults that present risk factors for cardiovascular diseases.

## 1. Introduction

Nitric oxide (NO), the main short-life vasodilatory endothelium-derived substance, is a low molecular weight gaseous compound (30.01 g·mol^−1^) containing 11 valence electrons and one unpaired electron, providing high NO reactivity and rapid oxidation to nitrite (NO_2_^−^) and nitrate (NO_3_^−^) [1]. This compound is involved in the central and peripheral nervous system neurotransmission, synapse plasticity mediation in nerve impulse transmission and inflammatory response, inhibiting platelet activation, adhesion and aggregation, and vascular tone regulation, diffusing across endothelial cells followed by vasodilation [2,3,4,5].

Unhealthy eating habits and lifestyles may stimulate the generation of oxygen (O_2_)-derived free radicals, contributing to an overall oxidative stress and inflammatory status. Indeed, obesity hypertension, diabetes mellitus, and hypercholesteremia, all associated risk factors for cardiovascular diseases (CVDs), may lead to an imbalance between the synthesis of endothelium-derived vasodilators and vasoconstrictors, resulting in diminished NO production and/or availability. This deleterious condition, known as endothelial dysfunction, leads to structural and functional changes in blood vessels, resulting in the exposure of the vascular lumen to a prothrombotic and fibrinolytic microenvironment, increasing arterial stiffness and creating favorable conditions for the development of atherosclerosis plaques and impairing vascular homeostasis and tone maintenance [6,7].

NO is synthetized during L-arginine metabolism, where this semi-essential amino acid serves as a substrate for a group of enzymes termed nitric oxide synthases (NOSs) (E.C. number: 1.14.13.39) [8,9]. Their three isoforms—neuronal nitric oxide synthase (nNOS or type I), inducible nitric oxide synthase (iNOS or type II), and endothelial nitric oxide synthase (eNOS or type III)—require certain cofactors, such as calmodulin, tetrahydrobiopterin, nicotinamide adenine dinucleotide phosphate (NADPH), flavin adenine dinucleotide (FAD), nicotinamine adenine dinucleotide (NAD), and molecular O_2_ [10,11]. However, oral L-arginine supplementation may not be sufficient to supply NOS with a substrate, since part of L-arginine is diverted to the gastrointestinal and hepatic systems during the synthesis of creatine and agmatine, and is also converted to L-ornithine and urea by arginase (E.C. number: 3.5.3.1) [11,12,13]. Alternatively, when orally administered, L-citrulline can be converted to L-arginine through alternative pathways, independent of the first-pass extraction, catalyzed by arginosuccinate lyase (E.C. number: 4.3.2.1) in the kidneys and consistently increasing L-arginine in plasma and in tissues, thus enhancing NO bioavailability [11,14].

Considering that endothelial dysfunction may be caused by a deficient L-arginine-NO metabolism, aggravated by the ineffectiveness of chronic L-arginine supplementation in promoting NO synthesis, this narrative review aims to summarize the potential benefits of natural but therapeutic L-citrulline administration through the ingestion of watermelon. The nutritional characteristics, bioactive compounds, and L-citrulline composition were detailed considering every edible part of watermelon. The L-citrulline metabolism, transport, and effects following watermelon supplementation on vascular and metabolic physiologies were addressed. Furthermore, the present review critically evaluated and analyzed the strength and limitations of a few clinical trials of L-citrulline effects supplementation, using watermelon as the source of this compound. Herein, a detailed agenda for future research on L-citrulline benefits was proposed, including long-term trials, dose–response studies, and investigations in specific populations.

## 2. Watermelon and Its Nutritional Composition

Watermelon originates from Equatorial Africa and has been cultivated for over 5000 years, being introduced in Brazil during the slave trade from around 1551 to 1557. This fruit belongs to the *Cucurbitaceae* family and *Citrullus* genus, which includes different species such as *Citrullus lanatus*, *Citrullus colocynthis*, *Citrullus ecirrhosus*, and *Citrullus rehmii* [15]. The size of watermelon fruit varies from less than 1 kg to more than 30 kg when cultivated in tropical climates, and can be round, oblong, cylindrical, or conical, with varied external rind colors. Watermelons consist of pulp, rind, and seeds, representing about 68%, 30%, and 2% of the total fruit weight compartments, respectively. The pulp can be white, yellow, orange, pink, or red, the latter the most common among the varieties traded in Brazil and in the Americas. The pulp color reflects the carotenoid composition, however it does not affect fruit flavor, which can vary according to the cultivar, ripening stage, and farming practices such as fertilization and irrigation, as well as phytosanitary plant status. The pulp can also vary in texture, with some cultivars having soft pulps and others firm or crunchy ones [15,16]. Furthermore, watermelons may contain around 200–800 seeds in different sizes, ranging from white, cream, green, red, and reddish-brown to brown and black, with or without spots [17,18] (Figure 1).

The nutritional composition of red watermelon and its byproducts is shown in Table 1. The seeds are considered the high-calorie part of watermelon (354–560 kcal·100 g^−1^ dwb) due to the high content of protein (31.99–49.70 g·100 g^−1^ dwb) and lipids (22.00–50.00 g·100 g^−1^ dwb). The fatty acid composition of watermelon seed oil has been reported as mainly comprising linoleic acid, followed by myristic, oleic, palmitic, stearic, and arachidic acids. Considering watermelon (*Citrullus lanatus*) seeds from fifteen cultivars regarding edible oil, high contents of monounsaturated fatty acid (MUFA) and polyunsaturated fatty acid (PUFA) have been detected, ranging from 11.06% to 20.04% and 43.95% to 55.29%, respectively [18,19].

The rind is considered a waste product, although recent reports point out the manufacture of watermelon rind-derivatives, such as candies, jam, and pickles [24,32]. Table 1 also shows that watermelon pulp and rind contain 90–92.5% and 67–90% (fwb) of moisture and watermelons are considered a low-calorie fruit (30–46 kcal·100 g^−1^ in pulp and ≈130 kcal·100 g^−1^ in rind fwb). The macronutrients that comprise watermelon pulp and rind include 0.20–0.25% and 0.10–0.83% lipids, 0.73–0.98% and 0.53–2.51% proteins, 4.23–11.60% and 4.50–25.50 carbohydrates, and 5.74–9.20% and ≈5.39% total sugars (fwb), responsible for watermelon’s sweetness.

Although the protein content is relatively low in watermelon pulp and rind and higher in seeds, this fruit contains essential and non-essential amino acids identified and quantified in their free forms [37]. It is important to note that L-citrulline is abundant in watermelon and can be progressively accumulated in both watermelon pulp and rind, particularly when farming under abiotic stress conditions, resulting in advantageous crops [16,37]. The L-citrulline content in red watermelon rind is between 60.0 and 500 mg·100 g^−1^ fwb (0.06–0.5 g·100 g^−1^ fwb), values higher than red watermelon pulp, ranging from 40.0 to 160 mg·100 g^−1^ fwb (0.04–0.16 g·100 g^−1^ fwb), and seeds (not detected) (Table 1). The L-citrulline contents in watermelon pulp from different cultivars are different, where yellow, orange, and red pulps contain 28.5, 14.2, and 7.9 mg·g^−1^ dwb or 0.28, 0.14, and 0.07 g·100 g^−1^ fwb (considering the 90% moisture of watermelon pulp) [38].

These findings are significant to achieve the minimum effective dose of L-citrulline (2–3 g·day^−1^) [39,40], it being necessary to consume < 1 to 3 kg·day^−1^ of fresh red watermelon rind and 2.5 to 5 kg·day^−1^ of fresh red watermelon pulp. The L-citrulline content in watermelon may be a limiting factor for its beneficial effect. However, the reduction in the volume or amount of water in watermelon byproducts to achieve adequate L-citrulline content may be used to concentrate L-citrulline content following the post-harvest processing of fresh watermelon and then contribute to the sustainability of fresh watermelon commercialization by promoting a bio circular economic mode [41,42]. Watermelon powder allows for a smaller concentration of L-citrulline because dehydration (water removal) concentrates nutrients and bioactive compounds per gram of product. Furthermore, the powdered form is more stable and easier to store and transport than the fresh fruit, and facilitates standardization, allowing for more precise control of the L-citrulline dose administered to volunteers in clinical and sports settings [20,33]. Due to its high nutritional value and remarkable sensory characteristics, watermelon juice generates great interest among consumers. Therefore, it is essential to use appropriate technologies that help maintain the sensory quality of the product while preserving its bioactive compounds, minimizing excessive consumption of the fruit and promoting adherence to supplementation regimens.

Regarding vitamin content in watermelon pulp, ascorbic acid (8.10–12.31 mg·100 g^−1^) and choline (4.10–4.50 mg·100 g^−1^) are found at the highest concentrations. In the watermelon rind, retinol activity equivalents (vitamin A) present the highest concentrations (50.15–52.13 mg·100 g^−1^), followed by ascorbic acid (5.35–8.46 mg·100 g^−1^), pyridoxin (5.34 mg·100 g^−1^), and riboflavin (0.02–2.71 mg·100 g^−1^). Due to oil presents in watermelon seeds, this part of watermelon contains highs tocopherol contents (37.50–53.00 mg·100 g^−1^), especially alpha-tocopherol, which is the most biologically active form (Table 2).

Regarding mineral content in watermelon pulp, potassium is found at the highest concentrations, ranging from 100.50 to 200 mg·100 g^−1^, followed by magnesium 10.00–15.00 mg·100 g^−1^, phosphorus 11.0–15.0 mg·100 g^−1^, and calcium 5.60–11.00 mg·100 g^−1^. In the watermelon rind and seeds, potassium is also found in higher concentrations, ranging from 21.70 to 447.33 and 482.30–1036.68 mg·100 g^−1^, followed by phosphorus 129.70–135.24 and 107.70–787.00 mg·100 g^−1^, calcium 28.00–29.15 and 57.20–758.20 mg·100 g^−1^, magnesium 1.48–35.00 and 86.20–515.00 mg·100 g^−1^, and sodium 11.40–12.65 and 7.10–99.00 mg·100 g^−1^. Furthermore, watermelon seeds contain high levels of copper and manganese (12.1–75.51 and 2.60–24.00 mg·100 g^−1^) (Table 2).

Watermelon also contains a wide range of other bioactive compounds with therapeutic importance, such as phenolic compounds, carotenoids, and cucurbitacin-derived compounds (Figure 1) [48,49]. Phenolic compounds are secondary metabolites generally found in plants and produced by the phenylpropanoid or polyketide pathways during plant growth or in response to abiotic stresses. These compounds contain one or more aromatic rings and at least one hydroxyl group, enabling them to bind to certain atoms and radicals, and are classified as simple phenols or complex phenols, according to the number of phenol units in their structures [49]. Phenolic compounds are the strongest antioxidant compounds capable of neutralizing reactive oxygen or nitrogen species, avoiding damage to macromolecules, such as the structural lipids, in cytoplasmatic and intracellular membranes and DNA, thus preventing several degenerative disorders [49,50].

Quantified by gallic acid equivalent (GAE), the seeds contain the highest total phenolic (4.20–4.50 mg of GAE·100 g^−1^) contents, followed by the rind (2.60–3.00 mg of GAE·100 g^−1^) and the pulp, which contains the lowest phenolic content (1.00–1.10 mg of GAE·100 g^−1^) (Table 2). Flavonoid compounds such as apigenin, myricetin, naringenin, luteolin, quercetin, rutin, and kaempferol; hydroxycinnamic acid derivatives like *p*-coumaric, m-coumaric, caffeic, sinapic, and ferulic acid; and hydroxybenzoic acid compounds such as 4-hydroxybenzoic, protocatechuic, gallic, and vanillic acid, have been detected in the watermelon matrix [51,52]. Meghwar et al. summarized in a table the main phenolic compounds found in each part of the watermelon. The phenolic compounds present in watermelon rind are gallic, synapic acid, hydroxycinnamic, quercetin, m-coumaric, chlorogenic, cyringic, *p*-coumaric, myricetin, caffeic, vanillic, and 4-hydroxybenzoic acid; those present in watermelon seeds are gallic, caggeic, syringic, sinapic, rosmarinic, vanillic, prochatechuic, leuteolin, chlorogenic, apigenic, 4-hydroxybenzoic, ferulic, and coumaric acid; and those present in watermelon pulp are carotenoids, quercetin, luteolin, gallic, coumarin and Aviprin [53].

Watermelon matrices also contain carotenoids, where the total carotenoids of red pulp correspond to 90–95% of lycopene and 5–10% of β-carotene contents, and lower phytofluene, phytoene, γ-carotene, ζ-carotene, and α-carotene concentrations. However, the total carotenoids of yellow pulp cultivar correspond to 30–40% of β-carotene, 15–25% α-carotene, and 10–20% lutein, and the white pulp cultivar contains only traces of carotenoids [54]. Red watermelon pulp contains more lycopene (4.40–8.00 mg·100 g^−1^) content when compared to the rind and seeds (Table 2). In red pulp watermelons, the lycopene content is very high, while in orange-pulp and yellow pulp watermelons, the lycopene content is very small.

Lycopene is a lipophilic red hydrocarbon pigment and carotenoid with a 40-carbon backbone composed of eight tetraterpene units and found mainly as cis-isomers in serum and tissues, more bioavailable forms than trans-isomers [55,56]. Lycopene intakes increase pharmacological benefits by acting as an antioxidant, neutralizing reactive species and improving the status of superoxide dismutase, catalase, and peroxidase, as well as nonenzymatic antioxidants like vitamins E and C in the cell [56,57]. Lycopene exhibits anti-inflammatory effects due to its lipophilic nature, enabling it to regulate inflammatory mediator signaling pathways and activate the expression of antioxidant genes in cell membranes. It can, for example, prevent the production of different cytokines, such as interleukins IL1, IL6, and IL8, tumor necrosis factor alpha (TNF-α), chemokines, and cyclooxygenase [58]. Furthermore, lycopene can inhibit injury caused by endoplasmic reticulum stress, prevent low density lipoproteins (LDLs) from oxidative damage, improve ventricular remodeling following myocardial infarction, reduce systolic blood pressure, enhance endothelial function, and play a chemoprotective role on benign prostate hyperplasia [59,60,61,62].

Red watermelon pulp contains more β-carotene (0.61–1.02 mg·100 g^−1^) content when compared to the rind and seeds (Table 2). β-carotene is a fat-soluble hydrocarbon carotenoid containing two cyclohexene types in both end groups (beta rings). It is formed by the cyclization of two terminal isoprene lycopene groups, and along with other carotenoids, is considered to be an antioxidant, providing protection against free radicals and oxidative damage by quenching singlet oxygens and scavenging peroxyl radicals. Furthermore, β-carotene acts as a retinol precursor, essential for normal growth and development, eyesight, and immune function, and protects human skin against infrared light-induced free radicals and mitigates type 2 diabetes mellitus (improves insulin and homeostatic model assessment for insulin resistance), obesity (inversely associated with the ratio waist-to-height, subcutaneous adipose tissue and visceral fat accumulation) and CVD (reduce triglyceride levels and increase NO production [63,64,65,66].

Xanthophylls are biosynthesized by carotenoid oxygenation, presenting similar structures of carotenes, but carotenes are purely hydrocarbons, which do not contain O_2_, while xanthophylls contain O_2_ atoms, making them more polar than carotenes. The hydroxylation of α-carotene produces α- cryptoxanthin, followed by lutein. The hydroxylation of β-carotene produces β-cryptoxanthin that, in turn, forms zeaxanthin, which is then epoxidated to violaxanthin and converted into neoxanthin. Neoxanthin, followed by violaxanthin and luteoxanthin, are the predominant xanthophylls found in yellow pulp watermelons. Zeaxanthin and lutein, for example, can absorb damaging blue and near-ultraviolet light in the human ocular globe, providing protection against macular degeneration [16]. Due to their characteristics as bioactive compounds, these phyto-compounds have been included as active and coloring additives by the food industry [67].

Cucurbitacins, a group of oxygenated steroidal triterpenes with a curcubitane skeleton, are synthesized via lanosterol, cycloarthenol, or parkeol, where the synthesis begins with 2,3-oxidosqualene cyclization to cucurbitadienol by oxidosqualene cyclase, followed by hydroxylation, acetylation, and glycosylation steps, producing cucurbitacin [68]. This compound is characterized by a tetracyclic cucurbitacin backbone, differing from tetracyclic triterpenes by the presence of numerous keto-, hydroxyl-, and acetoxy-groups. According to side-chain variations, cucurbitacins and their derivatives can be categorized into 12 main groups, namely A-T cucurbitacins [69]. Scientific evidence has demonstrated relevant biological and pharmacological activities, evidencing that cucurbitacins B, C, D, E, and I can inhibit the proliferation of certain cancer cell lines from breast, liver, pancreas, prostate, lung, central nervous system, and bladder carcinomas, as well as inhibiting acute and chronic leukemia cell line activities [70]. Furthermore, cucurbitacins B, D, E, and I inhibit cyclooxygenase-2 activity, impairing the conversion of arachidonic acid to prostaglandin H2, which is a common precursor for the synthesis of prostacyclin and thromboxane, both important inflammation and pain mediators [71]. Cucurbitacin R and dihydrocucurbitacin B can also inhibit cyclooxygenase-2 and TNF-α produced by activated macrophages and mast cells [71,72].

In addition, aldehydes and alcohols are part of these aroma volatile compounds accumulated during watermelon ripening, providing the peculiar watermelon flavor and aroma [73,74]. Although yellow watermelon and red watermelon share similar nutritional characteristics regarding water content, calories, carbohydrates, sugars, proteins, lipids, and fiber, without significant variations, they do present differences related to the levels of vitamins A and C, as well as carotenoids. The choice to emphasize the data related to red watermelon is due to its widespread production and consumption both in Brazil and abroad, whereas yellow watermelon occupies a smaller market share and is targeted at a more specific audience. Furthermore, the scientific literature has certain limitations, as most studies evaluating the composition of macro and micronutrients and bioactive compounds were performed on red watermelon, using uniform data and consolidated methods. However, this review included data on the main carotenoids found in yellow watermelon, highlighting its high content of β-carotene, α-carotene, and lutein, in contrast to the predominant lycopene in red watermelon.

## 3. L-Citrulline Metabolism and Transport

L-citrulline (C_6_H_13_N_3_O_3_) is a non-protein and non-essential solid and colorless α-amino acid at room temperature and standard pressure, with a melting point of 222 °C. As most other amino acids, it contains an asymmetric carbon, giving rise to two enantiomers, with the natural form comprising the L-isomer. The molecular weight of L-citrulline is 175.19 g·mol^−1^, and its chemical properties also result from the terminal ureide group of the aliphatic chain that replaces the α-carbon. L-citrulline is relatively soluble in water and slightly soluble in ethanol and methanol, due to its polar side chain. It contains two acidic groups from the carboxylic acid (pKa ≈ 2.4) and amine (pKa ≈ 9.4) [75].

The gastrointestinal absorption of exogenous L-citrulline following watermelon ingestion takes place between the median and lower ileum. L-citrulline is transported across enterocytes to the portal circulation mediated by sodium (Na^+^)-dependent transport, including the B^0,+^ amino acid transport system located in jejunum and ileum enterocytes, allowing for better L-citrulline bioavailability compared to L-arginine, not promoting osmotic diarrhea when administered at high doses. After its release from enterocytes, L-citrulline passes through the liver without undergoing any major metabolization, reaching the systemic circulation [76]. Despite this, the human liver can take up a substantial amount of intestinal-derived L-citrulline, thus influencing the amount of L-citrulline reaching the kidney, but the liver does not contribute significantly to circulating L-citrulline levels (Figure 2).

About 75–80% of the circulating L-citrulline extracted from the blood is transported to the kidneys and converted stoichiometrically into arginosuccinate in the presence of aspartate and adenosine triphosphate (ATP) by argininosuccinate synthetase (E.C. number: 6.3.4.5). Arginosuccinate is then converted into fumarate and L-arginine by arginosuccinate lyase (E.C. number: 4.3.2.1) as part of the urea cycle and L-arginine is then released into the renal vein and to the systemic blood circulation to be used by tissues [77]. The L-citrulline converted by the kidneys is enough to sustain whole-body L-arginine requirements. The biosynthesis of L-arginine from L-citrulline is the key mechanism for generating more bioavailable L-arginine, which takes place in the kidneys and represents 60% of the de novo whole-body L-arginine synthesis (Figure 2) [76,77].

L-citrulline can be generated endogenously during the amino acid metabolism of dietary-derived L-glutamine, L-proline, and L-arginine. L-glutamine contributes to 27.6% of all L-citrulline produced in the small intestine, where it is converted to glutamate by glutaminase (E.C. number: 3.5.1.2), which is then converted to ethyl pyrroline-5-carboxylate by pyrroline-5-carboxylate synthase (E.C. number: 2.7.2.11). Ethyl pyrroline-5-carboxylate is then converted to L-ornithine by ornithine aminotransferase (E.C. number: 2.6.1.13). L-ornithine is finally converted into circulating L-citrulline by ornithine transcarbamylase (E.C. number: 2.1.3.3) [11,78].

L-proline contributes 3.4% of the L-citrulline synthesis in enterocytes, where it is metabolized into pyrroline-5-carboxylate and, subsequently, into ornithine by proline oxidase (E.C. number: 1.5.99.8) and ornithine aminotransferase (E.C. number: 2.6.1.13). Finally, the ornithine produced in enterocytes from these amino acids is metabolized into L-citrulline by ornithine transcarbamylase (E.C. number: 2.1.3.3) [76,79].

L-arginine can also be converted into L-ornithine and urea, catalyzed by arginase (E.C. number: 3.5.3.1), an enzyme involved in the elimination of toxic nitrogen compounds. L-arginine is decarboxylated to agmatine by arginine decarboxylase (E.C. number: 4.1.1.19), which can be diverted to the synthesis of creatine by glycine amidinotransferase (E.C. number: 2.1.4.1). Furthermore, a portion of L-arginine from the renal metabolism reaches the bloodstream again and becomes a substrate for NOS enzyme isoforms, mainly eNOS, expressed in endothelial cells, to produce NO [79,80].

### 3.1. L-Citrulline Pharmacokinetics and Pharmacodynamics

Few studies have investigated the utilization of pharmacokinetics of ingested L-citrulline, demonstrating a correlated increase in endogenous L-arginine levels through L-citrulline supplementation although, in general, L-citrulline is able to form L-arginine with better performance considering time and plasmatic L-arginine levels. Oral L-citrulline administration rapidly increases circulating L-citrulline and L-arginine concentrations, peaking after 1.5 and 2 h and massively increasing in comparison to L-citrulline loads, returning to baseline within 5 and 8 h, respectively [11,81]. Plasmatic L-arginine concentrations were found to increase slowly and remain elevated for several hours following 2.0 g·day^−1^ of L-citrulline supplementation for 8 days in healthy men, whereas plasmatic L-arginine levels were not achieved or sustained after L-arginine ingestion at the same concentration [82]. Furthermore, different L-citrulline supplementation regimens at 0.75, 1.5, or 3 g, immediately improved (1.0 g) and sustained the L-arginine release when 1.6 g were administered twice a day for one week to middle-aged women and men. Oral L-citrulline dose-dependent supplementation increased areas under the plasma concentration–time curve and the maximal plasma concentration of plasmatic L-citrulline and L-arginine [11]. In addition, a significant increase in NO_3_^−^ and cyclic guanosine monophosphate (cGMP) urinary excretion were observed when 3 g of L-citrulline were administered. These findings strongly suggest that oral L-citrulline administration markedly increased de novo L-arginine synthesis, as well as the plasma concentrations of both amino acids and NO synthesis, to a greater extent than compared to only L-arginine supplementation under the same regimen [83].

It is important to note a lack of gastrointestinal distress following L-citrulline supplementation in all studies when compared to L-arginine supplementation. This is because oral L-citrulline administration bypasses the hepatic metabolism and is absorbed into small intestine enterocytes and from there, the portal circulation, by a distinct transport system comprising Na^+^-dependent transporters. Circulating L-citrulline, released from the gut, is absorbed by the proximal tubular cells of the kidney to comprise the urea cycle and de novo L-arginine synthesis to meet L-arginine body demands [14,76]. However, orally administered L-arginine is negatively affected by the gastrointestinal and first-pass hepatic metabolisms prior to system circulation, since high arginases levels in hepatocytes and enterocytes can rapidly hydrolyze L-arginine into L-ornithine and urea, reducing L-arginine availability in the bloodstream, which is subsequently transported by Na^+^-independent cationic amino acid transporters (CAT-1, 2, and 3) across the intestinal epithelium [76,77]. Furthermore, arginase levels, which have been reported to be elevated in hypertension, ischemic reperfusion injury, and vascular diabetes mellitus complications, compete directly with NOS for the L-arginine substrate. Increasing arginase synthesis and activity can reduce L-arginine availability, decrease NO production, and increase superoxide production, cell proliferation, and collagen formation, leading to vascular stiffness, a pathological condition usually concomitant with oxidative stress [84].

L-citrulline has been shown to display protective roles with regard to arginase increase and oxidative stress damage. Shatanawi and Momani demonstrated that plasma arginase activity was 1.7-fold higher in diabetic patients compared to healthy individuals [85]. Indeed, L-citrulline supplementation of 2 g·day^−1^ for one month caused a significant (21%) decrease in arginase activity in type 2 diabetes patients, also increasing plasmatic NO levels by 38%. The inhibitory effects on arginase also are evidenced after L-citrulline treatment in bovine and human endothelial cells displaying increased arginase activity [84,86]. Regarding oxidative stress management, 3 g of orally supplemented L-citrulline for 2 months decreased MDA and increased total antioxidant capacity in serum levels of diabetic individuals [86]. Based on these important experimental observations, L-citrulline can be recognized as an arginase activity inhibitor, reducing L-arginine hydrolysis and attenuating NOS competition, subsequently guaranteeing NO availability [84].

Clinical trials have been conducted to determine the dose ranging/tolerability for L-citrulline supplementation, as adverse gastrointestinal effects, such as nausea, vomiting, and diarrhea have been described following chronic or acute supplementations with L-arginine doses over 10 g in healthy and unhealthy volunteers. Intolerance to L-arginine is explained by rapid intestinal absorption saturation, with high loads inducing osmotic diarrhea [12]. In comparison, L-citrulline absorption is not a limiting factor to L-citrulline bioavailability, making supplementation more tolerable at doses over 15 g·day^−1^, not inducing gastrointestinal disturbances [12,14]. However, lower absorption rates and plasma L-citrulline retention have been reported in cases of 15 g L-citrulline oral intake. It seems that saturation of L-citrulline transporters may take place, although with no harmful gastrointestinal effects, leading to the conversion of renal L-citrulline to L-arginine [81]. Furthermore, doses over 3 g·day^−1^ are effective in increasing circulating L-arginine concentrations [14]. Therefore, the recommendation for the minimum effective dose of L-citrulline is ≈3 g·day^−1^, with a maximal effective dose of 12 g·day^−1^. Additionally, L-citrulline is a well-characterized and appropriate oral supplementation substance included in the list of bulk drug substances for oral administration by licensed pharmacists and physicians under section 503A of the Federal Food, Drug, and Cosmetic Act to treat urea cycle disorders [87,88,89].

### 3.2. L-Citrulline Regulatory Metabolism Roles

L-citrulline displays several regulatory physiological roles (Figure 3) [90,91,92,93]. In the liver, L-citrulline participates and accelerates ammonia clearance from the nitrogen of dietary proteins and muscle metabolism through various biochemical reactions, followed by ammonia metabolization and excretion as urea. As mentioned above, L-citrulline is converted into arginosuccinate by argininosuccinate synthetase, followed by arginosuccinate conversion into fumarate and L-arginine by arginosuccinate lyase. The cycle ends when arginase metabolizes L-arginine into L-ornithine, thus releasing urea. Finally, urea is taken up by the kidneys to be excreted, completing the urea cycle, due to shared metabolism between the liver and kidneys [77,94,95]. The liver L-citrulline metabolism is highly compartmentalized and is not connected with other L-citrulline pathways, since the hepatocytes involved in the urea cycle cannot take up L-citrulline from the portal circulation. Furthermore, ammonia is a caustic and corrosive compound and, at high concentrations formed during aerobic and anaerobic exercises, can result in harmful muscle cell effects, due to the inhibition of glycogen synthesis or breakdown, causing fatigue [40,95].

Several reports have highlighted the role of L-citrulline and branched-chain amino acids (BCAAs), such as L-leucine, L-valine, and L-isoleucine on increasing muscle protein synthesis by stimulating the translation initiation factor activity. This is important, especially during aging, because it is associated with a lower postprandial muscle protein synthesis rate due to higher splanchnic extraction in elderly subjects, meaning that fewer amino acids reach the systemic circulation, leading to the progressive loss of both muscle mass and function [76,94]. Despite presenting different chemical structures and metabolisms, both L-leucine and L-citrulline can stimulate muscle protein synthesis, as they are not metabolized by the liver, stimulating protein synthesis—L-leucine, in the postprandial state, and L-citrulline, in the post-absorptive state [96,97].

In the postprandial state, significant L-leucine provision stimulates insulin secretion and, in the presence of other essential amino acids, activates several intracellular signals involved in messenger ribonucleic acid (mRNA) translation initiation. These include the mammalian target for the rapamycin (mTOR) signaling pathway, which includes the 70 kDa ribosomal protein S6 kinase-1 and the eukaryotic initiation factor-binding protein 4E-1 [97,98]. In the post-absorptive state or under low protein intake diets, L-citrulline stimulates protein synthesis at a minimum level, due to its ability to induce the secretion of basal insulin and GH (growth hormone) to produce L-arginine, which in turn increases the mTOR signaling pathway. This favors the activation of eIF4E-eIF4G complexes in skeletal muscle and protein phosphorylation in the mTOR signaling pathway, particularly modulating both myofibrillar and sarcoplasmic protein syntheses [96,97].

L-citrulline is a natural L-arginine precursor, a semi-essential amino acid crucial in severe stresses, such as trauma, surgery, or infectious disease processes, as well as a substrate for NO synthesis [99,100,101]. When a portion of L-arginine from the renal L-citrulline metabolism reaches the bloodstream again, L-arginine can be transported to the intracellular environment through Na^+^-independent transport, mediated by specific CAT. Once in the intracellular environment, L-arginine activates the enzyme NOS and other cofactors such as calmodulin, tetrahydrobiopterin, NADPH, FAD, and NAD, and O_2_ catalyzes the five-electron O_2_-dependent oxidation of L-arginine, generating NO and L-citrulline. In addition, shear stress, the force produced by blood flowing at high velocity on the epithelial cells into the blood vessels, can activate NO synthesis through NOS [80].

Furthermore, large intestinal resection patients exhibit decreased absorption of L-citrulline, leading to intestinal failure, villous atrophy, and human immunodeficiency viral enteropathy [76]. Both L-citrulline and, consequently, L-arginine concentrations are decreased in this situation, with L-arginine supplementation seemingly logical. However, L-arginine undergoes significant liver uptake and supplementation with high L-arginine doses, leading to intestinal discomfort and diarrhea. The therapeutic use of L-citrulline, however, has been recommended due to its very high bioavailability [14,76]. One study carried out L-citrulline, instead of L-arginine, supplementation by the enteral route in an intestinal resection experiment, where rats were assigned to four groups: L-citrulline, L-arginine, control, and sham. The sham rats underwent transection, whereas the other three groups underwent 80% small intestine resection. Plasma and muscle L-arginine concentrations were higher in the L-citrulline group than in the other groups, while nitrogen balance was preserved in L-citrulline-treated resected rats but not in L-arginine-supplemented rats [102].

### 3.3. Nitric Oxide, the Final Cardiovascular Effector

NO is a gaseous and highly reactive molecule produced endogenously by L-arginine or L-citrulline and rapidly oxidized via autoxidation or reacting with the heme group of some proteins such as oxyhemoglobin, oxymyoglobin and copper transport plasma protein (ceruloplasmin) to produce NO_2_^−^ and subsequently, NO_3_^−^ [1,103]. Furthermore, NO can be synthetized by ingesting exogenous sources of NO_3_^−^, a nitric acid salt compound formed by a single nitrogen bonded to three O_2_ atoms, which is found in high contents (>1000 mg of NO_3_^−^·kg^−1^) in beetroot (*Beta vulgaris* subsp. *vulgaris*), leaf chicory (*Cichorium intybus*), beet greens (*Beta vulgaris* subsp. *vulgaris*), radish (*Raphanus raphanistrum* subsp. *sativus*), green spinach (*Spinacia oleracea*), and rocket (*Eruca vesicaria* subsp. *sativa*) [1,104].

Once synthetized and present in biological systems, NO is found in higher concentrations in lipophilic environments, embedded in membranes and in the hydrophobic regions of proteins. This compound has antioxidant properties through its free-radical scavenging ability to reduce reactive oxygen species, improves cardioprotective effects in the atherosclerotic process by preventing LDL cholesterol oxidation, and reduces reactive nitrogen species (RNS) production rates [104]. In immune cells, NO is produced during the inflammatory response by macrophages and other immune system cells, reacting with the superoxide anion and generating peroxynitrite, which, in turn, causes lethal damage to pathogens or tumoral cells by inactivating copper and iron-metalloproteins [5]. In synapse plasticity during the central and peripheral nervous impulse transmission, NO mediates and favors the secretion of neurotransmitters or hormones in neuronal junctions by diffusing to the presynaptic terminal and stimulating cGMP generation from the guanosine triphosphate (GTP) catalyzed by the soluble guanylate cyclase (sGC) enzyme. The cGMP acts on protein kinases, triggering the phosphorylation of target enzymes. Furthermore, the predominant mechanism that mediates NO on nervous system signaling involves the post-translational modification of Cys thiol residue nitrosylation, termed S-nitrosylation; Tyr nitration, termed 3-nitrotyrosination; and PKG-dependent Ser residue target protein phosphorylation [4,105].

Both endothelium- and platelet-derived NO prevent platelet aggregation and fibrin formation, inhibiting the spread of thrombus formation in blood vessels. NO performs its inhibitory action by reducing cytoplasmic calcium (Ca^2+^) by increasing Ca^2+^ extrusion and sarcoplasmic reticulum Ca^2+^-ATPase rates at the same time as decreasing Ca^2+^ inputs from the extracellular medium. NO increases the phosphorylation of the thromboxane-2 receptor and down-regulates P-selectin expression, preventing platelet activation and adhesion. In addition, NO modulates fibrinogen binding via the glycoprotein IIb and IIIa receptor, increasing the dissociation constant of this receptor to fibrinogen, reducing the total number of glycoprotein receptors on the platelet surface and resulting in unfavorable platelet aggregation conditions [106].

Under low O_2_ concentrations and pH, enzymes such as xanthine, aldehyde oxidases, aldehyde dehydrogenase type 2, carbonic anhydrase, or deoxyhemoglobin can reduce NO_2_^−^ to NO, which improves oxidative phosphorylation efficiency, evidenced by an increased P/O ratio, indicating no uncoupling mechanisms, such as proton leaks towards ATP synthesis and turnover, improving the ATP supply to skeletal muscle [104].

NO is noteworthy for its vascular tone regulation effects by crossing the endothelium and diffusing into the smooth muscle cells of blood vessels to form cGMP, which reduces intracellular Ca^2+^ concentrations by activating the Ca^2+^-pump within smooth muscle cells, promoting vasodilation through vascular tone reduction [107]. As smooth muscle contraction depends on the interaction between Ca^2+^ and calmodulin, reduced Ca^2+^ affects the formation of the Ca^2+^/calmodulin complex, causing vascular smooth muscle relaxation and vasodilation [108,109].

## 4. Cardioprotective Effects of Watermelon Ingestion

The CVDs, including atherosclerosis, have a complex pathophysiology characterized by a pro-inflammatory state with excessive production of free radicals—unstable, highly reactive molecules with unpaired electrons. This oxidative environment contributes to the dysregulation of plasma lipid levels, favoring the oxidation of LDL, forming oxy-LDL. Oxy-LDL accumulates in the subendothelial matrix, promoting the formation of foam cells and triggering endothelial dysfunction. During this process, NO synthesis, the main vasodilator produced by endothelial cells, is reduced, compromising vascular function [110]. Although not considered to exhibit proven biochemical synergy, carotenoids such as lycopene and β-carotene act as complementary antioxidants, neutralizing various free radicals [111,112].

In addition to being an alternative source of bioavailable NO through its L-citrulline content, the antioxidants from the watermelon matrix, such as polyphenols, lycopene, and vitamin C, are capable of neutralizing and removing reactive oxygen and nitrogen species, providing additional protection against cellular oxidative damage [16,21,53]. These antioxidants also restore eNOS activity and impaired NO bioavailability during oxidative stress by downregulating oxidants and inhibiting BH4 oxidation and eNOS uncoupling [113]. Finally, antioxidants such as lycopene have been found to inhibit endothelial cell migration, a critical step in angiogenesis, by modulating the endothelial growth factor-A signaling system, resulting in increased NO in endothelial cells [114]. For example, phenolic compounds and lycopene, owing to its highly conjugated structure, has a strong capacity to quench singlet oxygen (^1^O_2_), a highly reactive form of oxygen, by absorbing its energy and dissipating it as heat, thereby converting ^1^O_2_ into stable triplet oxygen. Lycopene also donates electrons to reactive oxygen species such as hydrogen peroxide, superoxide anion, and hydroxyl radicals, stabilizing them and preventing their interaction with NO and other biomolecules, thus preserving NO bioavailability [111,112]. Although β-carotene has fewer conjugated double bonds than lycopene, its structure is also effective against peroxyl and hydroxyl radicals by absorbing their energy and safely dissipating it, forming less reactive compounds [112,115]. Being lipid-soluble, both lycopene and β-carotene can incorporate into cell membranes and LDL, protecting these lipids from oxidation by peroxyl radicals. Furthermore, lycopene and β-carotene exhibit anti-inflammatory effects, modulating neutrophil and macrophage activity, influencing multiple stages of atherosclerosis, and contributing to improved endothelial function by favorably supporting NO action [111,112,115].

Evidence considering the cardioprotective effects mediated only by oral L-citrulline supplementation from watermelon has been shown in Table 3. Although L-citrulline supplementation effects have been supported by several studies, most clinical trials are conducted with the isolated amino acid, due to its non-toxic character and easy handling [116]. On the other hand, watermelon is the only significant dietary source of L-citrulline, and as it is part of the usual diet of many populations; also being a cheap option, it can be used as a usual and bioavailable source of L-arginine through L-citrulline conversion [38,116]. Because of the habitual consumption and good acceptance of this fruit, as well as its potential to provide L-citrulline, L-arginine, and NO, and thus achieve favorable cardiovascular outcomes in human beings, its ingestion can significantly contribute to public health.

A crossover pilot study conducted with pre-hypertensive adults demonstrated an encouraging result, because 6 weeks of watermelon supplementation (containing 2.7 g of L-citrulline and 1.3 g of L-arginine) reduced both central and peripheral pressures and arterial stiffness indices (*p* < 0.05) [117]. Similarly, a reduction in multiple blood pressure measurements and the carotid reflex index was observed after 6 weeks of supplementation with concentrated watermelon powder (providing 4 g of L-citrulline and 2 g of L-arginine), demonstrating its robust and clinically relevant physiological effect in obese and pre-hypertensive adults [118]. Treating pre-hypertensive obese individuals is relevant from a preventive point of view, because non-pharmacological interventions at this stage can delay or reduce the need for drug treatment. Thus, the combination of L-citrulline and L-arginine appears to improve central and peripheral hemodynamic pressures, due to increased NO bioavailability and vasodilation, which translates into improvements in measures relevant to reducing the cardiovascular risk in pre-hypertensive and obese volunteers. The use of concentrated watermelon powder allows for the provision of a greater quantity of L-citrulline and L-arginine in a smaller quantity of product, in addition to facilitating the standardization of the dose, compared to the consumption of fresh fruit, thereby improving the applicability in clinical studies and regular use as nutraceuticals for large populations.

However, Fan et al. [120] investigated the consumption of separate parts of watermelon rind (containing 387.4 mg of L-citrulline and 75.3 mg of L-arginine), flesh (containing 471.5 mg of L-citrulline and 175.2 mg of L-arginine), and seeds (containing 231.3 mg of L-citrulline and 69.3 mg of L-arginine) on the endothelial function of overweight/obese subjects. Following all supplementations of the watermelon samples, both plasma L-citrulline and L-arginine increased significantly (*p* < 0.05), but no effects on brachial artery flow-mediated dilation (FMD) were observed. Byproducts such as the rind and seeds are often discarded, but this study showed that they also contain significant amounts of L-citrulline and L-arginine, which led to a significant increase in the plasma levels of these compounds following their ingestion. However, the lack of improvement in FMD suggests that the functional response of the endothelium did not match the biochemical response, primarily due to a major weakness of the study: the small number of participants (only six) compromised the ability to detect subtle or moderate effects on endothelial function. Furthermore, there was no mention of variables such as the habitual diet and physical activity level of the individuals.

Despite the encouraging outcomes observed in selected trials, the current body of evidence remains insufficient to establish definitive clinical recommendations; however, these findings make it increasingly clear that watermelon supplementation holds promising therapeutic potential, especially for populations at high cardiovascular risk and risk of vascular dysfunctions, such as elderly and metabolic syndrome patients. The heightened sensitivity of these individuals to NO precursors such as L-citrulline and L-arginine suggests that their physiological systems may be more responsive to dietary interventions aimed at restoring endothelial balance and improving vascular tone. This responsiveness likely stems from a baseline deficiency or dysregulation in NO bioavailability in individuals presenting risk factors for the development of CVD, which creates a fertile ground for nutritional strategies to exert measurable effects. Therefore, the exclusion of physically active and healthy individuals from the observed benefits raises important questions about the interaction between baseline vascular health, lifestyle factors, and the efficacy of supplementation.

Subsequently, no changes in the level of serum L-citrulline and L-arginine and no hemodynamic effects were observed in healthy adults after 2 weeks of daily intake of 500 mL watermelon juice (containing 795 mg and 195 mg of L-citrulline and L-arginine, respectively) [121]. Similarly, no increases in plasma L-citrulline and L-arginine levels and no effects on FMD, pulse wave velocity (PWV), or blood pressure alterations were observed following the ingestion of 360 mL of watermelon juice (containing 1.63 g of L-citrulline and 1.15 g of L-arginine) by healthy postmenopausal women for four weeks [122]. The fact that there were no changes in the plasma levels of these amino acids or hemodynamic and vascular effects after weeks of consuming watermelon juice may seem disappointing, but the physiological context must be considered. In healthy individuals with no apparent vascular dysfunction, homeostatic regulation may have occurred that prevents significant increases in these amino acids in individuals without deficiency, and the endothelial system may be operating efficiently—which reduces the margin for noticeable improvements. Therefore, this study can be considered as a starting point rather than a conclusion. It suggests that watermelon, in all its parts, may be a viable source of functional amino acids, but we do not yet have enough evidence to translate it into measurable cardiovascular benefits. Studies with larger sample sizes, longer intervention periods, and controlled variables are needed.

On the other hand, Fujie et al. [123] reported a benefit on arterial stiffness evaluated by femoral-ankle PWV, alongside an increase in NO production and blood flow in the posterior tibial arteries after the administration of a single dose of wild watermelon-extracted juice (containing 162 mg of L-citrulline and 30 mg of L-arginine) to healthy females. In addition, Volino-Souza et al. [124] also reported similar results after ischemia–reperfusion injury induced in healthy individuals followed by a single intake of microencapsulated watermelon rind (containing 4 g of L-citrulline). It is important to emphasize that the study by Fujie et al. [123] did not show improvements in other hemodynamic and vascular parameters, and the study by Volino-Souza et al. [124] did not even analyze these parameters, guaranteeing conflicting results, because they demonstrated that the effectiveness of watermelon as a functional food is not universal, but contextual.

It is plausible that in individuals with optimal endothelial function—such as athletes or healthy adults—the physiological ceiling for NO-mediated improvements has already been reached, thereby limiting the observable impact of additional precursors. Furthermore, to obtain the maximum cardioprotective effect of L-citrulline intake from watermelon, the dosage and supplementation regimen must be considered. The variability in the studies, such as the design, sample size, watermelon formulation (juice, seeds, rind, microencapsulated products or whole fruit), and dosing regimens complicates the interpretation and generalization of the results. The distinction between acute and chronic supplementation also warrants attention. L-citrulline has a short half-life and an acute dose may temporarily elevate plasma levels, as shown in most of the studies mentioned above, but does not generate sustained beneficial effects on endothelial function, blood pressure, and arterial stiffness.

While single doses of watermelon-derived L-citrulline are sufficient to raise transient levels of this amino acid in plasma, future research should prioritize randomized, double-blind, placebo-controlled, and long-term modulation or ingestion (more than 2 months duration) studies, evaluating dose–response (minimum supplementation of ≈1–2 g of L-citrulline·day^−1^, increasing doses to 3 g–11 g of L-citrulline·day^−1^) to determine the optimal dose of L-citrulline extracted from watermelon and evaluate the biochemistry (inflammatory and NO production metabolic markers), hemodynamic, vascular, and endothelial function effects in sedentary elderly people (≥65 years old, with or without comorbidities), patients with moderate hypertension, and those presenting three or more risk factors for CVD (metabolic syndrome). In addition, different methods of administration of watermelon should be tested, such as concentrated watermelon rind juices, microencapsulated powder products from different parts of the watermelon, and the combination of L-citrulline from these watermelon products with other compounds, to verify the enhancement of the beneficial effects.

## 5. Conclusions and Future Perspectives

Nowadays, there is a significant growth in the demand for fruits, driven by factors such as the scarcity of raw materials in the food industry, population growth, and the preference for healthier eating habits. Given the current scenario, watermelon is a fruit which should be consumed either raw or processed and it is a significant source of bioactive substances displaying therapeutic importance, such as the content of phenolic compounds and carotenoids exerting biological and pharmacological properties such as antioxidant, anti-inflammatory, immunomodulatory and antimicrobial activities, inhibiting or bypassing the pathophysiological mechanisms involved in CVD and other age-related degenerative pathologies.

The presence of bioactive compounds with multiple biological functions in this food allows for its employment in the food industry, standing out as a strategic resource of great therapeutic usage. Scientific literature already includes several studies focusing on watermelon consumption. However, the amount of this fruit that must be ingested to achieve a level of bioactive compounds that could benefit individuals still represents a significant obstacle. Therefore, it is essential that innovative and technological research receives support to improve the efficiency of extracting or concentrating these bioactive components in watermelon.

Herein, we not only highlighted the potential of watermelon as a source of bioactive compounds with multiple biological functions, but also pointed out the paths for scientific and technological advancements in the employment of this fruit as a nutraceuticals source, primarily due to the presence of L-citrulline, an amino acid precursor for the most effective and potent synthesis of L-arginine, able to improve hemodynamic function, reinforcing watermelon’s role as an ally in the prevention and management of CVD. Oral administration of L-citrulline through watermelon requires additional evidence of its beneficial effects for individuals presenting CVD risk factors, but has demonstrated potential for efficiently improving hemodynamic function, since it increases plasma L-arginine concentrations, providing systemic L-arginine and not increasing L-arginine by the supplementation itself, probably because L-citrulline does not undergo hepatic metabolism, instead being transported to the kidneys to participate in the urea cycle or de novo L-arginine synthesis, subsequently increasing NO synthesis and promoting improved vasodilation and reducing blood pressure, peripheral arterial stiffness, and endothelial dysfunction in adults who already present risk factors for cardiovascular disease.

This review also revealed an important limitation of L-citrulline research to date concerning to the short-term nature of the clinical trials already performed. Furthermore, the direct role of L-citrulline beyond its function as an L-arginine precursor and the inter-action of L-citrulline with other pharmacological drugs commonly administered for the treatment of hypertension, atherosclerosis, insulin resistance, diabetes mellitus type 2, and CVD have not yet been well addressed. Therefore, the content of the present review guides a critical reflection on the role of science in the valorization of functional foods and the promotion of public health, highlighting watermelon as a promising and innovative source of nutraceutical and pharmacological compounds on the improvement of physio-pathological conditions that affect large populations.

## Figures and Tables

**Figure 1 nutrients-17-03221-f001:**
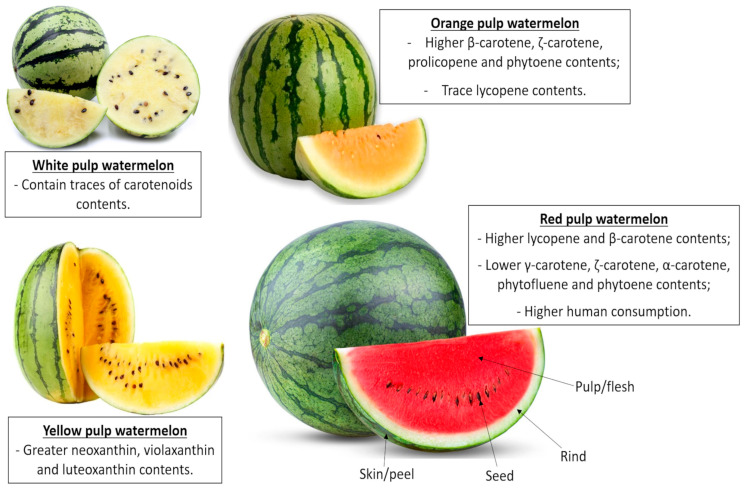
Transverse section of watermelon (*Citrulus lanatus*) cultivars and their main carotenoid contents.

**Figure 2 nutrients-17-03221-f002:**
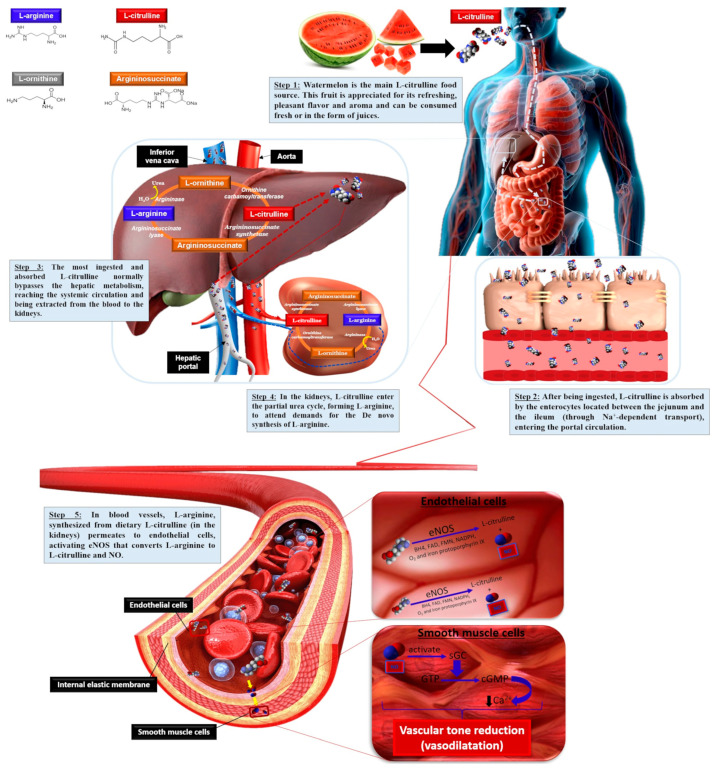
Oral L-citrulline via watermelon ingestion pathway. BH4, tetrahydrobiopterin; Ca^2+^, calcium; cGMP, cyclic guanosine monophosphate; eNOS, endothelial-nitric oxide synthase; FAD, flavin adenine dinucleotide; FMN, flavin mononucleotide; GTP, guanosine triphosphate; H_2_O, water; Na^+^, sodium; NADPH, nicotinamide adenine dinucleotide phosphate; NO, nitric oxide; O_2_, oxygen; sGC, soluble guanylate cyclase.

**Figure 3 nutrients-17-03221-f003:**
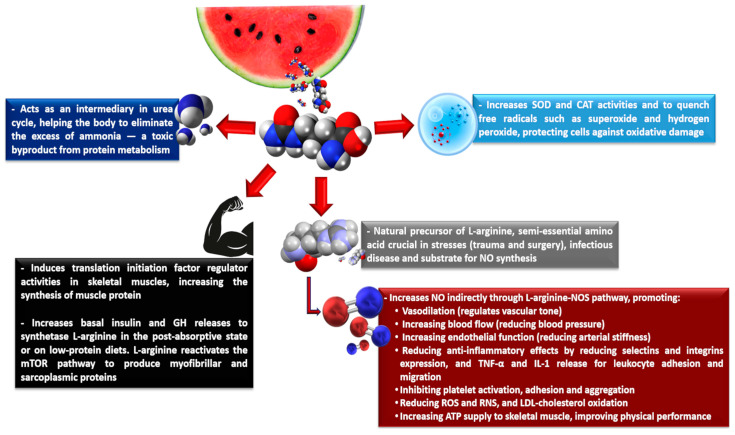
Brief representative scheme of the physiological L-citrulline functions. ATP, adenosine triphosphate; CAT, catalase; GH, growth hormone; IL-1, interleukin-1; LDL-cholesterol, low-density lipoprotein cholesterol; mTOR, mammalian target for the rapamycin; NO, nitric oxide; NOS, nitric oxide synthase; RNS, reactive nitrogen species; ROS, reactive oxygen species; SOD, superoxide dismutase; TNF-α, tumor necrosis factor alpha.

**Table 1 nutrients-17-03221-t001:** Proximate composition of red watermelon and its byproducts.

Compounds	Pulp/Flesh (fwb)	Rind (fwb)	Seeds (dwb)
Ashes (%)	0.25–0.50	1.00–3.84	2.30–5.10
Moisture (%)	85–95.2	65.00–85.00	5.05–10.06
Energy (kilocalorie)	30–46	130	354–560
Carbohydrate (%)	4.23–11.60	15.50–35.5	10.00–25.30
Total sugars (%)	5.74–9.20	5.39	3.23
Fructose (%)	2.72–4.11	0.20–0.75	ND
Glucose (%)	0.67–2.13	0.15–0.60	ND
Sucrose (%)	0.11–2.87	0.10–0.45	ND
Maltose (%)	0.02–0.14	tr	ND
Fiber (%)	0.2–0.73	3.00–5.59	14.50–20.10
L-citrulline (%)	0.04–0.16	0.06–0.5	ND
Total protein (%)	0.73–0.98	0.53–2.51	31.99–49.70
Glutamic acid (%)	0.063–0.080	ND	5.70–8.10
L-arginine (%)	0.055–0.075	0.054	5.00–7.00
Aspartate acid (%)	0.040–0.055	0.007	2.76–5.30
L-glycine	0.010–0.015	tr	1.20–2.50
L-phenylalanine (%)	0.015–0.022	0.003	2.03–3.00
L-valine (%)	0.015–0.022	0.003	1.56–2.10
L-serine (%)	0.018–0.025	0.018	1.51–2.1
L-threonine (%)	0.025–0.035	ND	1.11–1.80
L-leucine (%)	0.018–0.025	tr	2.15–2.80
L-isoleucine (%)	0.018–0.025	0.002	1.34–2.00
L-histidine (%)	0.010–0.015	tr	0.77–1.20
L-lysine (%)	0.055–0.075	tr	0.90–2.00
L-alanine (%)	0.018–0.025	0.007	1.48–2.11
L-methionine (%)	0.010–0.015	ND	0.83–2.17
L-tyrosine (%)	0.012–0.018	ND	1.30–1.60
L-proline (%)	0.022–0.030	0.01	1.25–2.00
Total lipids (%)	0.20–0.25	0.10–0.83	22.00–50.00
Myristic acid—C14:0 (%)	0.003–0.004	0.004	0.14–14.4
Palmitic acid—C16:0 (%)	0.0419–0.052	0.24	2.64–5.81
Margaric acid—C17:0 (%)	ND	ND	0.03–0.07
Stearic acid—C18:0 (%)	0.0118–0.014	0.045	1.90–6.55
Arachidic acid—C20:0 (%)	0.011–0.0125	0.005	0.1–1.19
Palmitoleic acid—C16:1 (%)	0.003–0.004	0.003	0.01–0.090
Oleic acid—C18:1 (%)	0.057–0.072	0.032	3.28–7.32
Gadoleic acid—C20:1 (%)	0.003–0.004	0.001	0.02–0.09
Linoleic acid—C18:2 (%)	0.048–0.060	0.252	13.14–28.1
Linolenic acid—C18:3 (%)	0.020–0.025	0.251	0.04–0.05

Values are expressed as means ± SD. (%) is the same as g·100 g^−1^. dwb, dry weight basis; fwb, fresh weight basis; ND, not detected; tr: trace level. Data are reported by: [16,20,21,22,23,24,25,26,27,28,29,30,31,32,33,34,35,36].

**Table 2 nutrients-17-03221-t002:** Vitamins, minerals, and bioactive compounds composition of red watermelon and its byproducts.

	Pulp/Flesh (fwb)	Rind (dwb)	Seeds (dwb)
**Vitamins (mg·100 g^−1^)**			
Retinol activity equivalents (A)	0.028	50.15–52.13	ND
Tocopherol (E)	0.03–0.04	0.02–0.04	37.50–53.00
Thiamine (B1)	0.03–0.05	0.03–1.23	0.13–2.20
Riboflavin (B2)	0.02–0.06	0.02–2.71	0.05–0.15
Niacin (B3)	0.18–0.3	0.04–4.25	0.33–3.55
Pantothenic acid (B5)	0.22	ND	0.27–0.35
Pyridoxin (B6)	0.05–0.07	5.34	0.1–3.2
Choline (B8)	4.10–4.50	ND	ND
Folate (B9)	ND	ND	0.3–0.10
Ascorbic acid (C)	8.10–12.31	5.35–8.46	4.21–6.81
**Minerals (mg·100 g^−1^)**			
Calcium (Ca)	5.60–11.00	28.00–29.15	57.20–758.20
Potassium (K)	100.50–200	21.70–447.33	482.30–1036.68
Magnesium (Mg)	10.00–15.00	1.48–35.00	86.20–515.00
Sodium (Na)	0.60–1.00	11.40–12.65	7.10–99.00
Iron (Fe)	0.19–1.0	1.30–4.63	4.50–8.40
Phosphorus (P)	11.0–15.0	129.70–135.24	107.70–787.00
Zinc (Zn)	0.1–1.5	1.29–5.10	4.10–10.40
Copper (Cu)	0.04–1.4	0.40–0.45	12.1–75.51
Selenium (Se)	ND	ND	ND
Manganese (Mn)	0.04–0.20	1.30–1.42	2.60–24.00
Cadmium (Cd)	ND	ND	ND
**Bioactive compounds (mg·100 g^−1^)**			
Lycopene	4.40–8.00	0.73–1.57	ND
β-carotene	0.61–1.02	0.12–0.65	ND
Total phenolic content (GAE)	1.00–1.10	2.60–3.00	4.20–4.50

Values are expressed as means ± SD. dwb, dry weight basis; fwb, fresh weight basis; ND, not detected. Data are reported by: [16,21,33,34,43,44,45,46,47].

**Table 3 nutrients-17-03221-t003:** Effects of watermelon supplementation on biochemical parameters, peripheral blood pressure, central hemodynamic parameters, and microvascular reactivity.

Watermelon (L-Citrulline Content)	Subjects/Age	Intervention Period	Clinical Trial Design	Main Outcomes	Reported by
Watermelon powder (2.7 g)	9 prehypertensive individuals (4 male/5 female) (54 y)	6 weeks	Crossover Placebo-controlled	↓ AIx, ↓ bPP, ↓ aSBP, ↓ aPP and ↓ P2 cfPWV—no effect	Figueroa et al. [117]
Watermelon powder (4 g)	14 obese prehypertensive or stage-1 hypertensive individuals (11 female/3 male) (58 y)	6 weeks	Randomized Placebo-controlled Two-periods crossover	↓ cAIx, ↓ SBP, ↓ DBP, ↓ MAP HR and ABI—no effects	Figueroa et al. [118]
Watermelon powder (4 g)	12 postmenopausal women (57 y)	6 weeks	Randomized Placebo-controlled Crossover	↑ plasma Cit, ↑ plasma Arg, ↓ baPWV, ↓ SBP, ↓ DBP and ↓ aortic SBP aortic and radial SBP—no effect	Figueroa et al. [119]
Fresh watermelon rind (387.4 mg) Fresh watermelon flesh (471.5 mg) Fresh watermelon seed (231.3 mg)	6 overweight/obese individuals (32.2 ± 7.6 y)	1 week	Randomized Placebo-controlled Crossover	FMD—no effect	Fan et al. [120]
Watermelon juice (795 mg)	17 healthy young adults (21–25 y) (6 males/11 females)	2 weeks	Randomized Placebo-controlled Double-blind Crossover	plasma Cit and Arg—no effects ↑ FMD	Vincellette et al. [121]
Watermelon juice (1.63 g)	21 healthy postmenopausal females (55–70 y)	4 weeks	Randomized Placebo-controlled Double-blind Crossover	plasma Cit and Arg—no effects FMD, PWV, MAP, SBP and DBP—no effects	Ellis et al. [122]
Watermelon juice (162 mg)	20 healthy females (20 y)	Single intake	Randomized Placebo-controlled Double-blind Crossover	↑ plasma NOx, ↑ tibial blood flow and ↓ faPWV SBP, DBP, baPWV, and cfPWV—no effects	Fujie et al. [123]
Microencapsulated watermelon rind (4 g)	12 healthy adults	Single intake	Randomized Placebo-controlled Single-blind Crossover	↑ plasma Cit, ↑ plasma Arg, ↑ plasma NOx and ↑ FMD	Volino-Souza et al. [124]

↑, increased; ↓, decreased; Arg, arginine; ABI, ankle-brachial index; AIx, augmentation index; aPP, aortic pulse pressure; aSBP, aortic systolic blood pressure; baPWV, brachial-ankle pulse wave velocity; bPP, brachial pulse pressure, cfPWV, carotid-femoral pulse wave velocity; cAIx, carotid augmentation index; Cit, L-citrulline; DBP, diastolic blood pressure; faPWV, femoral-ankle PWV; FMD, mediated flow dilatation; HR, heart rate; MAP, mean arterial pressure; NOx, nitrate and nitrite concentrations; PWV, pulse wave velocity; P2, amplitude of the second systolic peak; SBP, systolic blood pressure; y, years.

## Data Availability

Not applicable.

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
