# Peer review of "Watermelon Nutritional Composition with a Focus on L-Citrulline and Its Cardioprotective Health Effects—A Narrative Review"

_nutrients, 2025, doi:10.3390/nu17203221_

Round 1

Reviewer 1 Report

Comments and Suggestions for Authors

nutrients-3908673

  1. Merge overlapping discussions on watermelon composition (Sections 2 and 3) to avoid redundancy.
  2. While Table 3 is useful, the narrative should more critically evaluate the heterogeneity in clinical outcomes (e.g., dose-dependent effects, population-specific responses).
  3. The role of other watermelon bioactive compounds (e.g., polyphenols, carotenoids) in synergizing with L-citrulline to enhance NO bioavailability or reduce oxidative stress is underexplored. Expanding on these interactions would add novelty.
  4. Discuss how watermelon’s antioxidants (e.g., lycopene) may protect NO from oxidative degradation, enhancing its efficacy.
  5. Provide clear guidance on watermelon intake equivalents for achieving effective L-citrulline doses (e.g., grams of rind/pulp needed daily).

Author Response

Nutrients

 Manuscript ID: nutrients-3908673

Former Title: Nitric oxide synthesis through watermelon supplementation: L-citrulline, a novel cardiovascular compound, acting on the L-arginine-nitric oxide pathway in a synergistic way

Current Title: Watermelon nutritional composition as a source of L-citrulline, addressing its therapeutic potential in vascular and metabolic health: A Narrative Review

Special Issue: Benefits of Fruit Intake on Cardiovascular Health

Authors: Diego dos Santos Baião, Davi Vieira Teixeira Da Silva, Vania Margaret Flosi Paschoalin

Answers to Reviewer 1

GENERAL COMMENTS BY THE AUTHORS

We believe that we have fully addressed all reviewer concerns and comments.

Several modifications were carried out in the revised manuscript, as follows: 2 new references were added, discussions on watermelon composition were placed in topic 2 and the references were reorganized, the table 3 has been improved, sentence about synergy of bioactive compounds from watermelon to reduce oxidative stress and improving NO bioavailability was written, a guidance of L-citrulline ingestion doses from watermelon was included and the manuscript title has been changed. The entire text has been summarized reducing excessive paragraphs to turn it more concise. A list of abbreviations was included after the abstract section. All modifications were highlighted in yellow.

Modifications suggested by reviewers 1, 2 and 3 have polished the manuscript and increased its overall impact. We would like to thank reviewers 1, 2 and 3 for his/her insights and thoughtful critiques of our manuscript. By following the reviewer`s concerns, several points in the manuscript were better addressed and discussed, improving reader and understanding.

After performing the modifications suggested by the reviewers, the entire text was revised by an editing specialized company to improve English grammar and syntax.

Reviewer 1 comments precede our responses.

Comments and Suggestions for Authors

Comments to the Author

Merge overlapping discussions on watermelon composition (Sections 2 and 3) to avoid redundancy.

Answer: We agree with the reviewer and Topic 3, which more specifically describes and addresses the functions of the main bioactive compounds found in watermelon was shortened as recommended by Reviewer 2, and these informations was placed in Topic 2. "Watermelon and its nutritional composition." Furthermore, the reference in the text and the list of reference has been reorganize.

While Table 3 is useful, the narrative should more critically evaluate the heterogeneity in clinical outcomes (e.g., dose-dependent effects, population-specific responses).

Answer: We agree with the reviewer and the discussion of the articles in table 3 has been improved, considering all reviewers’ questions.

The role of other watermelon bioactive compounds (e.g., polyphenols, carotenoids) in synergizing with L-citrulline to enhance NO bioavailability or reduce oxidative stress is underexplored. Expanding on these interactions would add novelty.

Answer: As suggested by reviewers 2 and 3, the role of the main bioactive compounds found in watermelon was reduced. However, the sentence addressing the synergy of these bioactive compounds to reduce oxidative stress and improving NO bioavailability was written and added to the topic 4. Cardioprotective effects of watermelon ingestion, which had its name changed for a better understanding of the topic in general (page 16, lines 586-619).

Discuss how watermelon’s antioxidants (e.g., lycopene) may protect NO from oxidative degradation, enhancing its efficacy.

Answer: As suggested by the reviewer, a section emphasizing the protective role of antioxidants present in the watermelon matrix in restoring NO production and inhibiting its degradation was added to the manuscript (page 16, lines 597-605).

Provide clear guidance on watermelon intake equivalents for achieving effective L-citrulline doses (e.g., grams of rind/pulp needed daily).

Answer: We agree with the reviewer and a clear guidance of L-citrulline ingestion doses from watermelon rinds and pulps in grams was included (page 5, lines 146-163).

Reviewer 2 Report

Comments and Suggestions for Authors

This review examines the effects of L-citrulline on cardiovascular performance. The relevant comments are as follows:

Lines 82–85: Please check this passage, as the sentence appears to be repetitive.

Lines 79–87: Please review this paragraph. Starting every sentence with "Watermelon" makes the writing style monotonous.

Lines 232–233: Is the carotenoid content really that high? It cannot possibly account for the weight percentage of watermelon. In which nutrients do carotenoids constitute 99%?

The author extensively describes the value of lycopene, which is not the focus of this article. Are there any studies on extracting lycopene from watermelon? This would be more relevant than discussing the intrinsic value of lycopene itself.

Why does the table only include information on red watermelon? According to the manuscript, yellow watermelon and red watermelon are also important. If such information is available, please add it to the table.

The descriptions of polyphenols, lycopene, lutein, etc., are overly detailed. Please concisely summarize the characteristics of these active compounds derived from watermelon.

Line 160: There is a missing period at the end of this sentence.

The article appears to focus heavily on the nutritional value of watermelon and its active compounds rather than the metabolism of L-citrulline. Therefore, the author should consider revising the title to better reflect the current focus of the review.

Similarly, the section on NO and related content is excessively lengthy and unreasonable. The author should emphasize the observed metabolomic changes and underlying mechanisms.

Please indicate below the table what "↑" and "↓" represent.

The author provides detailed descriptions of certain studies, but the level of detail is excessive. It is unnecessary to include all research findings; instead, the author may reduce the description of target studies and focus more on presenting their own perspectives.

Conclusion: Please emphasize the guiding value of this review in the conclusion section.

Author Response

Nutrients

Manuscript ID: nutrients-3908673

Former Title: Nitric oxide synthesis through watermelon supplementation: L-citrulline, a novel cardiovascular compound, acting on the L-arginine-nitric oxide pathway in a synergistic way

Current Title: Watermelon nutritional composition as a source of L-citrulline, addressing its therapeutic potential in vascular and metabolic health: A Narrative Review

Special Issue: Benefits of Fruit Intake on Cardiovascular Health

Authors: Diego dos Santos Baião, Davi Vieira Teixeira Da Silva, Vania Margaret Flosi Paschoalin

Answers to Reviewer 2

GENERAL COMMENTS BY THE AUTHORS

We believe that we have fully addressed all reviewer concerns and comments.

Several modifications were carried out in the revised manuscript, as follows: sentences were rewritten for better understanding, at the topic “3. Bioactive watermelon compounds” and at the sub-section 4.3. Nitric oxide, the final cardiovascular effector had its text reduced, but a text addressing our own perspectives has been included in topic 5. Cardioprotective effects of L-citrulline supplementation through watermelon ingestion” and the manuscript title has been changed. The entire text has been summarized reducing excessive paragraphs to turn it more concise. A list of abbreviations was included after the abstract section. All modifications were highlighted in yellow.

Modifications suggested by reviewers 1, 2 and 3 have polished the manuscript and increased its overall impact. We would like to thank reviewers 1, 2 and 3 for his/her insights and thoughtful critiques of our manuscript. By following the reviewer`s concerns, several points in the manuscript were better addressed and discussed, improving reader and understanding.

After performing the modifications suggested by the reviewers, the entire text was revised by an editing specialized company to improve English grammar and syntax.

Reviewer 2 comments precede our responses.

Comments and Suggestions for Authors

Comments to the Author

This review examines the effects of L-citrulline on cardiovascular performance. The relevant comments are as follows:

Lines 82–85: Please check this passage, as the sentence appears to be repetitive.

Answer: The reviewer is correct, and the repetitive sentence has been removed from the manuscript, as suggested (page 2, lines 73-74).

Lines 79–87: Please review this paragraph. Starting every sentence with "Watermelon" makes the writing style monotonous.

Answer: The reviewer is correct and sentences were rewritten to improve the style, as suggested (page 2, lines 82-86).

Lines 232–233: Is the carotenoid content really that high? It cannot possibly account for the weight percentage of watermelon. In which nutrients do carotenoids constitute 99%?

Answer: The reviewer is correct since the sentence was written in a confusing way. The sentence was rewritten for better understanding, as suggested (page 7, lines 211-216).

The author extensively describes the value of lycopene, which is not the focus of this article. Are there any studies on extracting lycopene from watermelon? This would be more relevant than discussing the intrinsic value of lycopene itself.

Answer: We would like to clarify that the manuscript is not limited to describing the value of lycopene. Lycopene had been highlighted in the manuscript, because of it is the main carotenoid found in watermelon, and the text and table also addressed β-carotene, the second most important carotenoid found in red watermelon. In the topic 2, "Watermelon and its nutritional composition", the contents of lycopene and β-carotene contents are discussed very briefly in just one paragraph, due to their functional relevance. The mention to lycopene and β-carotene serves to justify their relevance within the manuscript context, without deviating from the main scope of this topic, which was to address the macronutrients, micronutrients, and bioactive compounds found in different fruit parts of the red watermelon. Furthermore, in the text, more specifically in topic 3, "Bioactive watermelon compounds", compounds such as xanthophylls and curcubitacins, in addition to β-carotene, are also mentioned to give a panel of the bioactive compounds in red watermelon.

Why does the table only include information on red watermelon? According to the manuscript, yellow watermelon and red watermelon are also important. If such information is available, please add it to the table.

Answer: We recognize the importance of expanding the results described in Table 1 by including data on macronutrients, micronutrients, and bioactive compounds for yellow watermelon. However, we chose to prioritize data for red watermelon because it is awidely cultivated and consumed nationally and internationally, while yellow watermelon represents a much smaller portion of the market, being restricted to specific niche this variety is widely cultivated and consumed nationally and internationally, while yellow watermelon represents a much smaller portion of the market, being restricted to specific niche.

This difference in magnitude justifies the analytical focus on the red variety, but both varieties have similar profiles in terms of water, calories, carbohydrates, sugars, proteins, lipids, and fiber, with no significant impact. The most significant differences occur in some micronutrients and bioactive compounds, such as vitamin A, vitamin C, and carotenoids. Furthermore, the manuscript includes yellow watermelon in the discussion of carotenoids, highlighting its rich composition in β-carotene, α-carotene, and lutein, in contrast to the lycopene predominant in the red variety. However, there are limitations in the scientific literature, since the studies majoritily used the red watermelon standardized data and consistent methodologies. The available data on yellow watermelon is scarcer and more heterogeneous, making it difficult to include it in a comparative table with the same level of rigor. A sentence explained this limitation was added in page 9, lines 292-303.

The descriptions of polyphenols, lycopene, lutein, etc., are overly detailed. Please concisely summarize the characteristics of these active compounds derived from watermelon.

Answer: We agree with the reviewer and Topic 3, which more specifically describes and addresses the functions of the main bioactive compounds found in watermelon was shortened as recommended by Reviewer 1, and these informations was placed in Topic 2. "Watermelon and its nutritional composition." Furthermore, the reference in the text and the list of reference has been reorganize

Line 160: There is a missing period at the end of this sentence.

Answer: The reviewer is correct and a period was added at the end of this sentence, as suggested.

The article appears to focus heavily on the nutritional value of watermelon and its active compounds rather than the metabolism of L-citrulline. Therefore, the author should consider revising the title to better reflect the current focus of the review.

Answer: The reviewer is correct, and the manuscript title was changed to meet the review concern “Watermelon nutritional composition as a source of L-citrulline, addressing its therapeutic potential in vascular and metabolic health: A Narrative Review”

Similarly, the section on NO and related content is excessively lengthy and unreasonable. The author should emphasize the observed metabolomic changes and underlying mechanisms.

Answer: The reviewer is correct and the topic 4.3. Nitric oxide, the final cardiovascular effector has been summarized, emphasizing the metabolomic changes and underlying mechanisms of NO, as suggested.

Please indicate below the table what "↑" and "↓" represent.

Answer: The reviewer is correct and symbols were indicated in the legend of table 3, as suggested (page 18).

The author provides detailed descriptions of certain studies, but the level of detail is excessive. It is unnecessary to include all research findings; instead, the author may reduce the description of target studies and focus more on presenting their own perspectives.

Answer: The reviewer is correct, excessive description of findings were excluded and our own perspectives has been included in topic 4, as suggested.

Conclusion: Please emphasize the guiding value of this review in the conclusion section.

Answer: The reviewer is correct and the conclusion has been changed, emphasizing the guiding value of the present review (page 20-21), as suggested.

Reviewer 3 Report

Comments and Suggestions for Authors

The manuscript provides a broad and detailed review of L-citrulline metabolism, nitric oxide synthesis, and the potential cardiovascular benefits of watermelon supplementation. It is well organized, comprehensive, and addresses a timely topic of relevance to nutrition and cardiovascular health. However, the paper in its current form requires major revisions before it can be considered for publication. The discussion relies heavily on descriptive summaries rather than a critical synthesis of the available clinical evidence. Most cited studies are short-term and small in scale, yet the manuscript often draws broad conclusions without sufficient caution. Furthermore, the direct role of L-citrulline beyond its function as an L-arginine precursor, as well as possible interactions with commonly used cardiovascular or metabolic drugs, are underdeveloped. The sustainability and by-product valorization aspects are interesting but need deeper integration with the main narrative. I recommend major revisions to (1) critically evaluate the strength and limitations of existing clinical trials, (2) expand the discussion on mechanisms beyond L-arginine conversion, (3) provide more balanced conclusions, and (4) clearly outline specific priorities for future research.

  1. Critically evaluate the strength and limitations of existing clinical trials, avoiding overgeneralization from short-term or small-sample studies.

  2. Expand the discussion of L-citrulline’s mechanisms beyond its role as an L-arginine precursor, including arginase inhibition, oxidative stress modulation, and muscle protein synthesis.

  3. Include a section on potential drug–nutrient interactions with antihypertensive, antidiabetic, and anti-atherosclerotic therapies.

  4. Revise conclusions to be more cautious and balanced, clearly separating established evidence from speculative findings.

  5. Better integrate sustainability and by-product valorization into the main narrative rather than treating them as detached topics.

  6. Provide a more detailed agenda for future research, including long-term trials, dose-response studies, and investigations in specific populations such as elderly or metabolic syndrome patients.

  7. Improve language clarity by simplifying overly long sentences and correcting occasional grammatical inconsistencies.

  8. Ensure uniformity in citation style and verify that all references are up to date and correctly formatted.

  9. Tables and figures could be more concise — consider summarizing key data rather than presenting overly detailed values.

  10. Standardize terminology (e.g., always use “L-citrulline” instead of alternating with “citrulline”).

  11. Add abbreviations list or ensure all abbreviations are defined at first use (e.g., NOS, NO, BCAA).

  12. Review the abstract to reduce length and highlight the most important findings more succinctly.

  13. Consider adding a graphical summary or schematic figure to improve accessibility for non-specialist readers.

Author Response

Nutrients

Manuscript ID: nutrients-3908673

Former Title: Nitric oxide synthesis through watermelon supplementation: L-citrulline, a novel cardiovascular compound, acting on the L-arginine-nitric oxide pathway in a synergistic way

Current Title: Watermelon nutritional composition as a source of L-citrulline, addressing its therapeutic potential in vascular and metabolic health: A Narrative Review

Special Issue: Benefits of Fruit Intake on Cardiovascular Health

Authors: Diego dos Santos Baião, Davi Vieira Teixeira Da Silva, Vania Margaret Flosi Paschoalin

Answers to Reviewer 3

GENERAL COMMENTS BY THE AUTHORS

We believe that we have fully addressed all reviewer concerns and comments.

Several modifications were carried out in the revised manuscript, as follows: sentences were rewritten for better understanding, at the topic “3. Bioactive watermelon compounds” and at the sub-section 4.3. Nitric oxide, the final cardiovascular effector had its text reduced, but a text addressing our own perspectives has been included in topic 5. Cardioprotective effects of L-citrulline supplementation through watermelon ingestion” and the manuscript title has been changed. The entire text has been summarized reducing excessive paragraphs to turn it more concise. A list of abbreviations was included after the abstract section. All modifications were highlighted in yellow.

Modifications suggested by reviewers 1, 2 and 3 have polished the manuscript and increased its overall impact. We would like to thank reviewers 1, 2 and 3 for his/her insights and thoughtful critiques of our manuscript. By following the reviewer`s concerns, several points in the manuscript were better addressed and discussed, improving reader and understanding.

After performing the modifications suggested by the reviewers, the entire text was revised by an editing specialized company to improve English grammar and syntax.

Reviewer 3 comments precede our responses.

Comments and Suggestions for Authors

Comments to the Author

The manuscript provides a broad and detailed review of L-citrulline metabolism, nitric oxide synthesis, and the potential cardiovascular benefits of watermelon supplementation. It is well organized, comprehensive, and addresses a timely topic of relevance to nutrition and cardiovascular health. However, the paper in its current form requires major revisions before it can be considered for publication. The discussion relies heavily on descriptive summaries rather than a critical synthesis of the available clinical evidence. Most cited studies are short-term and small in scale, yet the manuscript often draws broad conclusions without sufficient caution. Furthermore, the direct role of L-citrulline beyond its function as an L-arginine precursor, as well as possible interactions with commonly used cardiovascular or metabolic drugs, are underdeveloped. The sustainability and by-product valorization aspects are interesting but need deeper integration with the main narrative. I recommend major revisions to (1) critically evaluate the strength and limitations of existing clinical trials, (2) expand the discussion on mechanisms beyond L-arginine conversion, (3) provide more balanced conclusions, and (4) clearly outline specific priorities for future research.

Answer: The reviewer is correct and these recommendations has been included in the manuscript.

Critically evaluate the strength and limitations of existing clinical trials, avoiding overgeneralization from short-term or small-sample studies.

Answer: As shown in the abstract and objectives on page 2 (lines 71-76), this narrative review aimed to summarize the potential benefits of therapeutical L-citrulline administration by watermelon ingestion, not by the intake of the isolated natural compound or by synthetic L-citrulline. The studies mentioned in the manuscript were critically evaluated. The studies addressing the L-citrulline intake through watermelon presented limitations such as short-term trials, small samples size, and lack of sample standardization. Therefore, at the end of the discussion, a more detailed agenda for future research has been included, as recommended by Reviewer 3.

Expand the discussion of L-citrulline’s mechanisms beyond its role as an L-arginine precursor, including arginase inhibition, oxidative stress modulation, and muscle protein synthesis.

As suggested by the reviewer, the biological effects of L-citrulline goes beyond its bioconversion into L-arginine and NO production. Previous studies have demonstrated the role of L-citrulline in protein synthesis, which is important, for example, during aging, when the loss of muscle mass and function is increased. Furthermore, the biological effects of L-citrulline on factors that compromise endothelial function, such as inhibiting arginase activity, attenuating oxidative stress, and improving antioxidant status, have been demonstrated in endothelial cell cultures and in clinical trials. Considerations of these points are described on pages 12-13, lines 430-446; page 13, lines 481-490; and page 14, lines 496-501, as suggested by Reviewer 3.

Include a section on potential drug–nutrient interactions with antihypertensive, antidiabetic, and anti-atherosclerotic therapies.

Answer: We appreciate the reviewer's suggestion and consider it very relevant. However, literature specifically addressing the effects of L-citrulline interactions concomitantly with drug therapies for diabetes, hypertension, and other cardiovascular health-related medications is very scarce. This lack of interactions between L-citrulline and hypertensive drugs was addressed in Page Furthermore, when specifically focused on watermelon intake, which is the focus of this review, there is a lack of data to justify a section addressing drug-nutrient interactions. Therefore, we believe that including the suggested section is neither possible nor pertinent. Although the literature addressing L-citrulline interactions to conventional drug therapies for diabetes, hypertension, and other cardiovascular health-related medications is very scarce, a sentence including drug-nutrient interactions were added – Page 22, lines 772- 781, to rise this relevant point, as suggested by the reviewer.

Revise conclusions to be more cautious and balanced, clearly separating established evidence from speculative findings.

Answer: The reviewer is correct, and conclusion section has been revised, as suggested (page 20-21).

Better integrate sustainability and by-product valorization into the main narrative rather than treating them as detached topics.

Answer: We agree with the reviewer and accepted his-her suggestion. Topic 3, which describes and addresses the functions of the main bioactive compounds found in watermelon was shortened as recommended by Reviewer 1 and 2, and these information were replaced in Topic 2 "Watermelon and its nutritional composition." Furthermore, the reference in the text and the list of references were reorganized.

Provide a more detailed agenda for future research, including long-term trials, dose-response studies, and investigations in specific populations such as elderly or metabolic syndrome patients.

Answer: The reviewer is correct, and a sentence with these information has been included in page 20, lines 709-734, as suggested

Improve language clarity by simplifying overly long sentences and correcting occasional grammatical inconsistencies.

Answer: The reviewer is correct, and the entire text was revised by an editing specialized company to improve English grammar and syntax, as suggested.

Ensure uniformity in citation style and verify that all references are up to date and correctly formatted.

Answer: The reviewer is correct, and all references list were reviewed (in the text and in the list of reference), as suggested.

Tables and figures could be more concise — consider summarizing key data rather than presenting overly detailed values.

Answer: The reviewer is correct, and the key data of tables have been summarized. Table 1 and table 2 were compressed, reducing the spacing between lines, as suggested.

Standardize terminology (e.g., always use “L-citrulline” instead of alternating with “citrulline”).

Answer: The reviewer is correct and the terminology L-citrulline has been standardized in the entire manuscript, as suggested.

Add abbreviations list or ensure all abbreviations are defined at first use (e.g., NOS, NO, BCAA).

Answer: The reviewer is correct and abbreviations were defined at first use in the manuscript, as suggested.

Review the abstract to reduce length and highlight the most important findings more succinctly.

Answer: The reviewer is correct, and the abstract was shortened, by highlighting the most important findings succinctly (page 1), as suggested.

Consider adding a graphical summary or schematic figure to improve accessibility for non-specialist readers.

Answer: A graphical abstract was added to the manuscript, as suggested.

Round 2

Reviewer 1 Report

Comments and Suggestions for Authors

All questions have been addressed in the revised manuscript. 

Author Response

Nutrients

 Manuscript ID: nutrients-3908673

Former Title: Nitric oxide synthesis through watermelon supplementation: L-citrulline, a novel cardiovascular compound, acting on the L-arginine-nitric oxide pathway in a synergistic way

Current Title: Watermelon nutritional composition as a source of L-citrulline, addressing its therapeutic potential in vascular and metabolic health: A Narrative Review

Special Issue: Benefits of Fruit Intake on Cardiovascular Health

Authors: Diego dos Santos Baião, Davi Vieira Teixeira Da Silva, Vania Margaret Flosi Paschoalin

Answers to Reviewer 1

GENERAL COMMENTS BY THE AUTHORS

Modifications suggested by reviewers 1, 2 and 3 have polished the manuscript and increased its overall impact. We would like to thank reviewers 1, 2 and 3 for his/her insights and thoughtful critiques of our manuscript. By following the reviewer`s concerns, several points in the manuscript were better addressed and discussed, improving reader and understanding. After performing the modifications suggested by the reviewers, the entire text was revised by an editing specialized company to improve English grammar and syntax.

Reviewer 2 Report

Comments and Suggestions for Authors

Approved

Author Response

Nutrients

 Manuscript ID: nutrients-3908673

Former Title: Nitric oxide synthesis through watermelon supplementation: L-citrulline, a novel cardiovascular compound, acting on the L-arginine-nitric oxide pathway in a synergistic way

Current Title: Watermelon nutritional composition as a source of L-citrulline, addressing its therapeutic potential in vascular and metabolic health: A Narrative Review

Special Issue: Benefits of Fruit Intake on Cardiovascular Health

Authors: Diego dos Santos Baião, Davi Vieira Teixeira Da Silva, Vania Margaret Flosi Paschoalin

Answers to Reviewer 2

GENERAL COMMENTS BY THE AUTHORS

Modifications suggested by reviewers 1, 2 and 3 have polished the manuscript and increased its overall impact. We would like to thank reviewers 1, 2 and 3 for his/her insights and thoughtful critiques of our manuscript. By following the reviewer`s concerns, several points in the manuscript were better addressed and discussed, improving reader and understanding. After performing the modifications suggested by the reviewers, the entire text was revised by an editing specialized company to improve English grammar and syntax.

Reviewer 3 Report

Comments and Suggestions for Authors

This is a comprehensive and scientifically grounded narrative review on Citrullus lanatus and its L-citrulline–mediated vascular and metabolic benefits. The manuscript is well organized and relevant; however, it requires refinement to emphasize novelty, reduce descriptive redundancy, moderate causal language, and strengthen the clinical evidence synthesis. 

  • The review is comprehensive but overly descriptive; please emphasize novel insights or unique perspectives that distinguish this paper from previous reviews on L-citrulline and vascular health.

  • Streamline long compositional sections (e.g., taxonomy, seed oil details) to maintain focus on cardiovascular and metabolic mechanisms.

  • The causal language (“promotes,” “confers benefits”) should be moderated to reflect the mainly associative nature of the available clinical evidence.

  • Consider adding a summary table of human clinical studies (dose, duration, outcomes) to clarify the strength and consistency of findings.

  • The discussion and conclusion should include clearer guidance for future research—particularly standardization of L-citrulline intake, long-term efficacy, and potential antioxidant synergy.

Author Response

Nutrients

 Manuscript ID: nutrients-3908673

Former Title: Nitric oxide synthesis through watermelon supplementation: L-citrulline, a novel cardiovascular compound, acting on the L-arginine-nitric oxide pathway in a synergistic way

Current Title: Watermelon nutritional composition as a source of L-citrulline, addressing its therapeutic potential in vascular and metabolic health: A Narrative Review

Special Issue: Benefits of Fruit Intake on Cardiovascular Health

Authors: Diego dos Santos Baião, Davi Vieira Teixeira Da Silva, Vania Margaret Flosi Paschoalin

Answers to Reviewer 3

GENERAL COMMENTS BY THE AUTHORS

We believe that we have fully addressed all reviewer concerns and comments.

Several modifications were carried out in the revised manuscript, as follows: a refinement to emphasize novelty and to reduce descriptive redundancy was realized, emphasized novel insights or unique perspectives that distinguish this paper from previous reviews, topic 2. Watermelon and its nutritional composition was reduced, a summary table of human clinical studies (Table 3) was added and clearer guidance for future research was added to the manuscript. The entire text has been summarized reducing excessive paragraphs to turn it more concise. A list of abbreviations was included after the abstract section. All modifications were highlighted in yellow.

Modifications suggested by reviewers 1, 2 and 3 have polished the manuscript and increased its overall impact. We would like to thank reviewers 1, 2 and 3 for his/her insights and thoughtful critiques of our manuscript. By following the reviewer`s concerns, several points in the manuscript were better addressed and discussed, improving reader and understanding.

After performing the modifications suggested by the reviewers, the entire text was revised by an editing specialized company to improve English grammar and syntax.

Reviewer 3 comments precede our responses.

Comments and Suggestions for Authors

Comments to the Author

This is a comprehensive and scientifically grounded narrative review on Citrullus lanatus and its L-citrulline–mediated vascular and metabolic benefits. The manuscript is well organized and relevant; however, it requires refinement to emphasize novelty, reduce descriptive redundancy, moderate causal language, and strengthen the clinical evidence synthesis.

Answer: We agree with the reviewer and a refinement to emphasize novelty and to reduce descriptive redundancy were realized in the entire text.

The review is comprehensive but overly descriptive; please emphasize novel insights or unique perspectives that distinguish this paper from previous reviews on L-citrulline and vascular health.

Answer: We agree with the reviewer and novel insights or unique perspectives that distinguish this paper from previous reviews about L-citrulline was added in the abstract and in the page 2, lines 74–83.

Streamline long compositional sections (e.g., taxonomy, seed oil details) to maintain focus on cardiovascular and metabolic mechanisms.

Answer: We agree with the reviewer and this section (topic 2. Watermelon and its nutritional composition) had already been shortened as recommended by Reviewer 1 and 2. We have reduced this section a little further (highlighted in yellow), but some informations had to be kept, as they were suggestions from reviewers 1 and 2.

The causal language (“promotes,” “confers benefits”) should be moderated to reflect the mainly associative nature of the available clinical evidence.

Answer: We agree with the reviewer and causal language was reduced in the entire manuscript.

Consider adding a summary table of human clinical studies (dose, duration, outcomes) to clarify the strength and consistency of findings.

Answer: We agree with the reviewer and a Table 3 was added to the manuscript, as suggested (page 17).

The discussion and conclusion should include clearer guidance for future research—particularly standardization of L-citrulline intake and long-term efficacy.

Answer: We agree with the reviewer and clearer guidance for future research was added to the manuscript, as suggested (page 19, lines 659-684, and page 19-20, lines 699-728).
